# Examining area-level variation in service organisation and delivery across the breadth of primary healthcare. Usefulness of measures constructed from routine data

D. C. Butler[1]*, L. R. Jorm[2], S. Larkins[3], J. Humphreys[4], J. Desborough[1], K. J. Korda[1]

1 Research School of Population Health, Australian National University, Canberra, Australia, 2 Centre for Big Data Research in Health, University of New South Wales, Kensington, Australia, 3 Anton Breinl Research Centre for Health Systems Strengthening, James Cook University, Townsville, Australia, 4 School of Rural Health, Monash University, Melbourne, Australia

* Danielle.Butler@anu.edu.au

**Data Availability Statement:** ABS Census and PHIDU data are available for download at https://

## Abstract

### Background

Australia has a universal healthcare system, yet organisation and delivery of primary healthcare (PHC) services varies across local areas. Understanding the nature and extent of this variation is essential to improve quality of care and health equity, but this has been hampered by a lack of suitable measures across the breadth of effective PHC systems. Using a suite of measures constructed at the area-level, this study explored their application in assessing area-level variation in PHC organisation and delivery.

### Methods

Routinely collected data from New South Wales, Australia were used to construct 13 small area-level measures of PHC service organisation and delivery that best approximated access (availability, affordability, accommodation) comprehensiveness and coordination. Regression analyses and pairwise Pearson's correlations were used to examine variation by area, and by remoteness and area disadvantage.

### Results

PHC service delivery varied geographically at the small-area level–within cities and more remote locations. Areas in major cities were more accessible (all measures), while in remote areas, services were more comprehensive and coordinated. In disadvantaged areas of major cities, there were fewer GPs (most disadvantaged quintile 0.9[SD 0.1] vs least 1.0[SD 0.2]), services were more affordable (97.4%[1.6] bulk-billed vs 75.7[11.3]), a greater proportion were after-hours (10.3%[3.0] vs 6.2[2.9]) and for chronic disease care (28%[3.4] vs 17.6 [8.0]) but fewer for preventive care (50.7%[3.8] had cervical screening vs 62.5[4.9]). Patterns were similar in regional locations, other than disadvantaged areas had less after-hours

www.abs.gov.au/websitedbs/censushome.nsf/
home/historicaldata2006 and https://phidu.torrens.
edu.au/social-health-atlases respectively. Other
data used in this study cannot be shared publicly
because data are owned by a third party and
authors do not have permission to share these
data. Detailed information for accessing the data
underlying the results presented in the study are
available from: AIHW health workforce, https://
www.aihw.gov.au/about-our-data/accessing-data-
through-the-aihw/data-on-request; AMPCo Medical
workforce, https://www.ampco.com.au/ampco-
data-services/; HERO data, http://www.healthstats.
nsw.gov.au; MBS claims, https://www.
servicesaustralia.gov.au/organisations/about-us/
reports-and-statistics/statistical-information-and-
data#contacts; and ARIA + https://arts.adelaide.
edu.au/hugo-centre/services/aria.

**Funding:** DB. This research was supported through
a grant from the Australian Government through
the National Health and Medical Research Council
Postgraduate Scholarship (GNT1038903). The
funders had no role in study design, data collection
and analysis, decision to publish, or preparation of
the manuscript.

**Competing interests:** The authors have declared
that no competing interests exist.

care (1.3%[0.7] vs 6.1%[3.9]). Measures were positively correlated, except GP supply and affordability in major cities (-0.41, p < .01).

## Implications

Application of constructed measures revealed inequity in PHC service delivery amenable to policy intervention. Initiatives should consider the maldistribution of GPs not only by remoteness but also by area disadvantage. Avenues for improvement in disadvantaged areas include preventative care across all regions and after-hours care in regional locations.

## Introduction

Measuring how organisation and delivery of primary health care (PHC) services varies between and within countries is essential to identify best practice for achieving system goals of high-quality care and health equity [1–4]. While Australia has universal health care, PHC service organisation and access to care varies according to remoteness [5–8] and area disadvantage [7–10], with the disparity in access between metropolitan and more remote areas pre-eminent in the literature and policy discourse [11–14]. However, the nature and extent to which PHC service organisation and delivery in Australia varies according to local context is not well characterised. The ability to do this has been hampered by a lack of suitable measures that encompass the breadth of effective PHC systems.

Conceptual models characterise different levels of PHC operation [1, 2, 4, 15]. The structural level relates to national policies (e.g. on universal access or workforce). The service delivery level (or process level) relating to organisational characteristics of health services more generally (access, including availability, affordability and accommodation) and those considered integral to primary healthcare (comprehensiveness, continuity and coordination). These PHC service delivery characteristics can be viewed to operate, in part, at the geographic area-level, and are modifiable. Hence, they provide potential avenues to reduce unwarranted variation in PHC use and quality of care.

Opportunities for action requires appropriate empirical measurement of the organisation and delivery of PHC services of areas. Currently there is inconsistency in how service delivery characteristics are measured [1–4], and the empirical evidence for each specific measure is highly variable internationally, and largely absent in Australia. Using data from 2009–2013, recent studies have examined the structure and organisation of PHC systems across European countries [2, 16, 17], and in the largest study to date also included Australia, New Zealand and Canada (Quality and Costs in Primary Care in Europe, QUALICOPC) [18–21]. These data have been used to investigate PHC service characteristics with respect to variation between countries and practices [16, 22], the relationship with quality of care [19, 23] and health outcomes [20, 24]. Many of these studies have been based on composite scores across several indicators for each dimension of PHC service delivery (that is, access, comprehensiveness, and coordination) rather than examining the specific measures themselves [2, 16, 23]. While this is helpful in comparing health systems across countries, it is not useful for identifying a specific dimension of PHC service delivery within a country requiring policy or practice intervention. Later studies, using country-level measures linked to individual survey data, have focussed on service characteristics at the practice- or provider/patient-level [17, 19–22]. Variation in characteristics may differ at the area-level requiring an alternative policy approach. Further, while there are some Australian survey data [22], the overall sample size was small without adequate geographic spread limiting the assessment of how PHC service delivery varies across areas.

A range of Australian routine data sources can potentially be used to measure PHC service characteristics at the area-level across the dimensions of access, comprehensiveness and coordination [6, 8, 25–27]. To date, there is limited information on these potential measures, and their geographical variation and how this relates to remoteness and area socioeconomic disadvantage [5–7, 28]. This information is critical for policy and service delivery action, as there may be aspects of the PHC system at the area-level that could be modified to reduce unwarranted variation in PHC use and quality of care. The current study addressed this knowledge gap by constructing, from routinely collected data, a suite of area-level measures of PHC service organisation and delivery in Australia, covering access, comprehensiveness, and coordination, to explore their utility in assessing area-level variation in the PHC system.

## Methods

### Defining PHC service characteristics

This study examined three area-level dimensions of PHC service delivery: access, comprehensiveness, and coordination. For the purposes of this study access was defined as availability (the supply of primary care providers for the population), affordability (financial barriers to receiving services) and accommodation (organisation of services to accept clients, such as hours of operations, mode of service provision [face to face or telehealth]) [16, 17, 29]. Comprehensiveness referred to care provided for most needs (chronic disease, acute, preventative, maternal health and so on) and coordination to the ability of primary care providers to coordinate use of other services (specialist and allied health), skill mix and diversity within the service, and team-based care [2, 15, 17]. Where possible we constructed variables that approximated different aspects of each dimension to examine how PHC service organisation and delivery may vary between areas.

Acceptability (a feature of access) and continuity are considered integral service characteristics of effective PHC systems [2, 17, 29]. However, both relate more to the practice/provider and patient interface, rather than one amenable to change at the area-level and as such were not considered further for the study.

### Data and study setting

PHC measures were based on routine data available at the small-area level across the state of New South Wales (NSW), the most populous state in Australia, comprising 32% of the Australian population (N = 7,232,589) [30]. The data covered the period 2006 to 2012.

Data relating to primary care workforce, location of community health services, sociodemographic characteristics and disease prevalence were obtained from the Australian Medical Publishing company (AMPCo) doctor mailing lists (2010); Australian Institute of Health and Welfare (AIHW) health workforce survey (2007); Public Health Information Development Unit (PHIDU) Social Health Atlas of Australia (2010, 2011, 2014); Health Establishment Registration Online (HERO) data, Ministry of Health NSW 2012 and Australian Bureau of Statistics (ABS) 2006 Census.

Data relating to PHC services delivered were obtained from Medicare Benefit Schedule (MBS) claims data 2008–2012. The MBS database includes claims for subsidised medical and diagnostic services provided by registered medical and other practitioners through Medicare (Australia's universal health insurer [31]), including all out-of-hospital general practice and specialist attendances, as well as diagnostic and therapeutic procedures. Aggregated data were provided by MBS item groups of interest by specific request (S1 Appendix for item numbers and groups included). Data on the proportion of MBS claims for services provided without an

associated out-of-pocket cost for individuals (referred to as bulk-billed) were obtained from National Health Performance Authority (NHPA), 2011–2012.

## Variables constructed and data aggregation

A total of thirteen variables were constructed; four relating to availability, four relating to affordability, one relating to accommodation and four capturing comprehensiveness and two capturing coordination (Table 1). These variables were:

**Access.** General practitioner (GP, Australia's primary care providers) availability measured as number of GPs, full-time equivalent (FTE) and full-time work-load equivalent (FWE); affordability measured by i) out-of-pocket (OOP) expenses, ii) the proportion of GP services bulk-billed, iii) percentage eligible persons health care card holders, and iv) percentage eligible persons pension card holders. Health care cards or pension cards (typically for the those with a disability, unemployed persons, low-income earners or the elderly), are Government funded equity measures intended to minimise cost barriers to medical services and pharmaceuticals; accommodation measured by percentage after-hours care services (of total GP services).

Pairwise correlation analysis between availability variables revealed estimates derived from MBS claims (PHIDU FWE*)* did not correlate with other measures of availability and were negatively associated in major cities (S1 Appendix). Further, AMPCo data has been shown to overestimate actual numbers of GPs in more remote areas (28). Hence, only the AIHW variable was retained for the main analysis. The percentage of health care and pension card holders within an area was strongly associated with area disadvantage (adjusted R2 in bivariate analyses 0.8 and 0.6 respectively). These measures were most likely a proxy indicator for area socioeconomic position, rather than capturing an aspect of PHC affordability within an area. As such, these two measures were not included in the main analysis.

**Comprehensiveness.** Two measures of preventative care: i) percentage eligible women who participated in cervical screening (for the majority of Australian women, this is provided in primary care), and ii) MBS-funded health assessment services claimed per 100 eligible persons. These health assessments involve an assessment by a GP of a patient's physical, psychological and social wellbeing.

**Comprehensiveness and coordination.** Two measures capturing both these dimensions of PHC: i) number of chronic disease care (CD care) MBS services claimed per 100 self-reported long-term conditions, these services include chronic disease care planning as well as coordination and referral to other providers to enact that care plan; and ii) number of community health centres relative to the population of that area, as typically, these centres in NSW encompass coordination and comprehensiveness in their models of care.

All area PHC service characteristic measures were calculated at the Statistical Area Level 3 (SA3) (populations of between 30,000 and 130,000 persons). Numerators and denominators for each variable represent a full census count for that area, except for long-term conditions, which were modelled estimates of the number of people who self-reported a long-term condition, available by broad condition group only from National Health Survey data [32]. Given counts are available by broad group only, totals for each SA3 represent the total number of conditions within that area. Data not available at SA3 level were re-assigned using relevant population weighted correspondences publicly available through the ABS (that is, from 2006 statistical geographies to 2011 statistical geographies).

Pairwise correlation demonstrated that variables within dimensions (that is, between variables measuring a common underlying construct) of the area PHC service characteristics were correlated (S1 Appendix). Affordability variables—OOP expenses and bulk-billing—were

**Table 1. Area-level PHC service characteristic measures, data sources and geographical aggregation.**

| Dimension | Variable name | Measure | Data source | Aggregation |
|---|---|---|---|---|
| **Access** | | | | |
| Availability | *AMPCo count* | Headcount per 1000 URP | AMPCo doctor list, 2010. Denominator URP Census 2006 | SLA2006 to SA3 2011. Denominator CCD2006 to SA3 2011 |
| | *AMPCo FTE* | Full time/part time equivalent per 1000 URP | AMPCo Doctor list (part time assigned 0.5 FTE), 2010. | SLA2006 to SA3 2011. Denominator CCD2006 to SA3 2011 |
| | | | Denominator URP Census 2006 | |
| | *AIHW FTE* | Full time equivalent (hours) per 1000 URP | AIHW Health Workforce survey, 2007 | SLA2006 to SA3 2011. Denominator CCD2006 to SA3 2011 |
| | | | Denominator URP Census 2006 | |
| | *PHIDU FWE* | Full time work load (services) equivalent per 1000 URP | MBS-derived, published by PHIDU (years 2009–2010), Social Health Atlas 2011 | SLA2006 to SA3 2011. Denominator CCD2006 to SA3 2011 |
| | | | Denominator URP Census 2006 | |
| Affordability | *OOP expenses* | Out-of-pocket costs per service in dollar amounts | GP MBS service claims for the years 2009–2011, DHS | Provided at SA3 |
| | *Bulk-billing* | Percentage of all non-referred GP attendances that did not attract a co-payment | MBS service claims for the years 2011–2012, supplied to NHPA by DHS | Provided at SA3 |
| | *HCC holders* | Percent of 15–65 year olds with a HCC (2009) | Centrelink, June 2009; Denominator, ABS ERP June 2008. Compiled by PHIDU 2010 | SLA2006 to SA3 2011 |
| | *PC holders* | Percentage of over 15 year olds with a pension card (2009) | Centrelink, June 2009; Denominator, ABS ERP June 2008. Compiled by PHIDU 2010 | SLA2006 to SA3 2011 |
| Accommodation | *After-hours care* | Percentage of after-hours care and after-hours urgent care items claimed of total GP services | GP MBS services claimed for the years 2009–2011, DHS | Provided at SA3 |
| **Comprehensiveness** | | | | |
| Preventative care | *Cervical screening* | Percentage of 20–69 year old women who participated in cervical screening in the last 2 years | NSW cancer registry data for 2011–2012; Denominator, ERP from ABS 2011–2012 (denominator). Variable compiled by PHIDU 2014 | SLA2006 to SA3 2011 |
| | *Health assessments* | Number of items claimed for health assessments per 100 eligible persons (3–5 year olds, 45–49 year olds, 75 years and over and all Aboriginal and Torres Strait Islander peoples) | GP MBS service claims for the years 2000–2011, DHS; ABS 2006 census | Provided at SA3; denominator CCD2006 to SA3 2011 |
| **Comprehensiveness and coordination** | | | | |
| Chronic Disease Care | *CD care* | Number of items claimed for GP management plans, team care arrangements and cycles of care for diabetes and asthma per 100 long-term conditions | GP MBS service claims for the years 2009–2011, DHS; denominator, self-reported long-term conditions compiled by PHIDU 2011 from NHS 2007–2008, and ERP average of June 2007–2008 | Provided at SA3; denominator aggregated from SLA2006 to SA3 2011 |
| Community Health Centre | *Community Heath centres* | Number of community health centres per 10,000 population | HERO Data, Ministry of Health, NSW. | Geocoded from address to SA3 |
| | | | Denominator URP Census 2006 | |

Abbrev. URP, usual resident population; ERP, estimated resident population; AMPCo; Australian Medical Publishing Company; AIHW, Australian Institute of Health and Welfare; PHIDU, Public Health Information Development Unit; FTE, full-time equivalent; FWE, full-time workload equivalent; SA3, Statistical Areas Level 3; SLA, statistical local area; CCD, census collection district; NHS, national health survey; DHS, Department of Human Services; ARIA, accessibility and remoteness index of Australia; HC, health centre; NSW, New South Wales; SE, socioeconomic; IRSD, index of relative socioeconomic disadvantage; SEIFA, socioeconomic indexes for small areas; HERO; health establishment registration online; NHPA; National Health Performance Authority; GP; general practitioner; MBS, Medicare Benefits Scheme; GISCA; National Centre for Social Applications of Geographical Information Systems. URP is used as denominator population for numerator data from census years. ERP is used as denominator population for numerator data from inter-census years.

highly correlated (-0.93, p < .001). There were moderate correlations between comprehensiveness and coordination variables (health assessments and CD care, 0.53, p < .001), while CD care was negatively associated with cervical screening (-0.28, p < .01).

## Analysis

All area-level primary health care service characteristics were examined as categorical and continuous variables. Categorical variables were constructed using the ranked quartile population distribution breakpoints across SA3s.

To describe the geographic variation of service characteristics within NSW, means and standard deviations and median and interquartile ranges for continuous variables, and proportions for categorical variables, were calculated where appropriate; together by small area (SA3), separately by region and area disadvantage, and by disadvantage within regions. Region was categorised as major cities, inner regional or outer regional/remote, based on the Australian Statistical Geography Standard Classification. Area disadvantage was assigned based on population-weighted quintiles of the ABS Index of Relative Socio-Economic Disadvantage [33] (1 = most disadvantaged, 5 = least disadvantaged) [34]. Linear regression was performed for test of trend. Categorical variables were also mapped by SA3 to describe the geographical distribution of the PHC service characteristics across the state.

To examine how dimensions of PHC service delivery relate to each other at the area-level in the Australian context, pairwise Pearson's correlation analyses were performed between variables from each dimension (e.g. affordability and comprehensiveness). Correlation coefficients and significance tests are reported. Chi squared tests for overall trend between categorical variables were also performed.

Analyses were performed used Stata 12 and the GRAPHC online mapping tool [35]. Approval for this project was obtained from the Australian National University Human Research Ethics Committee (protocol 2011/703) and the AIHW Ethics Committee (EC 2010/2/23).

## Results

### Geographic variation in PHC service characteristics and relationship with remoteness and area disadvantage

Most of the SA3s in NSW were located in major cities (44%) with the majority of the population residing in these urban areas (58.4%), with an overall median population of 65,142 (IQR 63,713). Areas that were more disadvantaged were also more remote (p < .001). Geographic variation of PHC service characteristics was evident for all measures, across all dimensions (Fig 1, S1 Appendix). For example, per capita GP FTE varied from 0.6 per 1000 persons in areas with the least GPs (34.1% of SA3s) to 1.1 in areas with the most GPs (23.1% of all SA3s).

Measures of PHC service organisation and delivery related to access, comprehensiveness and coordination varied according to region and area disadvantage. (Table 2, Fig 2). Areas in major cities had the most GP FTEs per capita, (mean = 0.9[SD: 0.2], vs outer regional/remote mean = 0.7[SD: 0.2]), were the most affordable (% GP attendances bulk-billed: 87[11.5] vs 80.4 [7.9]) and had the most after-hours care (% GP attendances: 7.2[2.9] vs 1.4[1.8]). Outer regional/remote areas had the most health assessments claims (per 100 eligible persons: 11 [4.2] vs major cities 8.5[2.2]) and community health centres (per 10,000 population: 11.4[6.5] vs 6.1[3.8]). The most disadvantaged areas were the most affordable when considering both OOP expenses (AUD2.7[2.6] vs least disadvantaged AUD6.0[4.8]) and bulk-billing (88.2% [9.7] vs 77.4%[20.9]) and had the highest rates on most measures of comprehensiveness (CD care items per 100 eligible population: 26[6.7] vs 15.9[9.3]; health assessments per 100 eligible population: 10.3[4.0] vs 7.7[4.0]) and coordinated care (community health centres per 10,000 population: 9.3[6.5] vs 6.7[4.9]). Areas that were the least disadvantaged had the highest rates of cervical screening participation (per 100 eligible persons: 62.2[4.7] vs 52.3[3.6]). Similar relationships were observed with categorical variables.

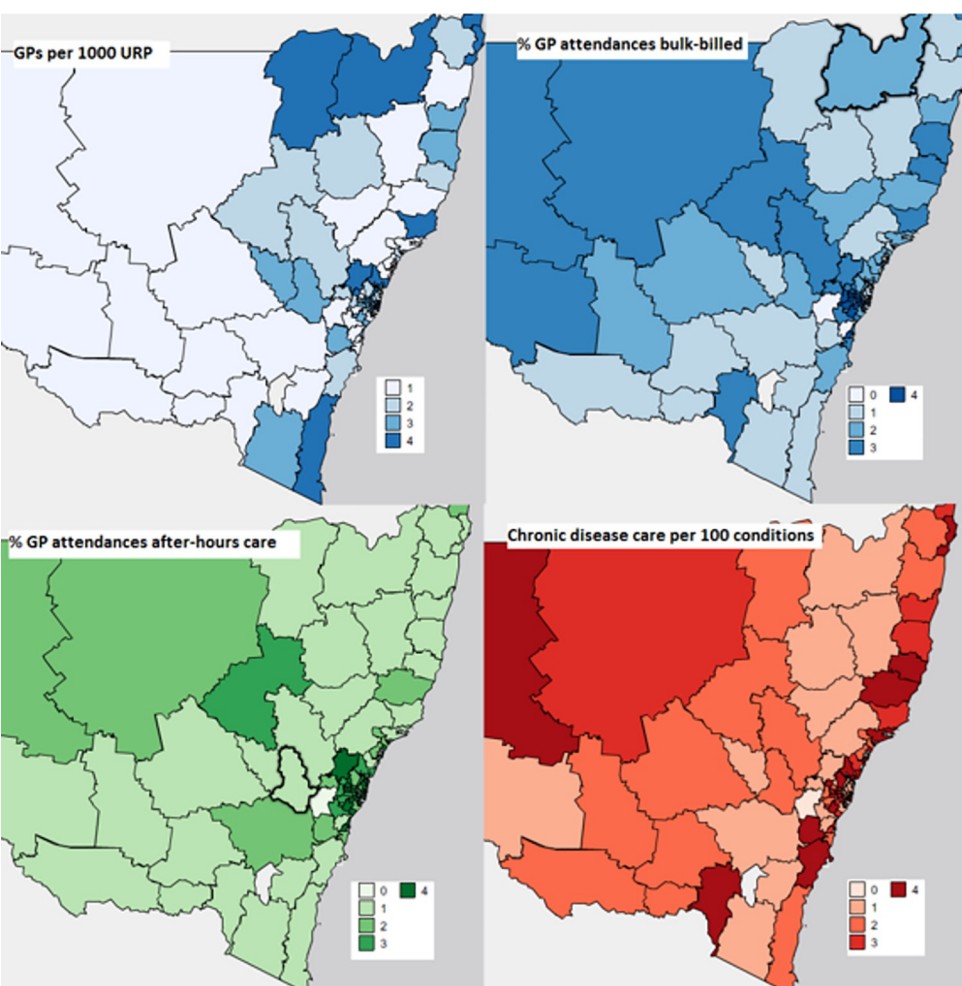

**Fig 1. Geographical distribution of area PHC service characteristic quartiles from each domain and subdomain of PHC systems, by SA3.** Abbrev. PHC, primary health care; SA3, Statistical Area Level 3; GP, general practitioner; URP, usual resident population; %, percent. Legend gives population-weighted quartile categories for each area PHC characteristic, where 1 is the first and lowest quartile and 4 is the fourth and highest quartile. Zero (0) refers to SA3s for which data were missing.

When stratified by region, the relationship of characteristics with area disadvantage was generally similar to the overall trend (Table 3); for example, disadvantaged areas in major cities were the most affordable, had the most chronic disease care and least cervical screening. Additionally, it was found that disadvantaged areas in major cities had fewer GPs (FTE per capita 0.9[0.1] vs 1.0[0.2]). For some measures, trends varied according to region, and in a few cases the direction of association reversed. For example, in contrast to major cities, rates of after-hours care in inner regional areas were lower in the most disadvantaged areas (most disadvantaged compared with least: major cities 10.3%[3.0] vs 6.2%[2.9]; inner regional 1.3%[0.7] vs 6.1%[3.9]).

## Association between dimensions of PHC service characteristics at the area-level

Areas with a greater presence of a PHC service characteristic from one domain of either access (or components of access), comprehensiveness or coordination were more likely to have

**Table 2. Area PHC characteristics by region and area SE disadvantage, mean (SD).**

| | Availability | Affordability | | Accommodation | Comprehensiveness and Coordination | | | |
|---|---|---|---|---|---|---|---|---|
| Region | GP FTE per capita | OOP expenses | Bulk-billing | After-hours care | Cervical screening | HA | CD care | Community health centres |
| Major cities | 0.9(0.2) | 3.5(4.2) | 87(11.5) | 7.2(2.9) | 55.7(6.4) | 8.5(2.2) | 23.8(8.2) | 6.1(3.8) |
| Inner regional | 0.8(0.2) | 4.6(2.9) | 79.4(9.3) | 3.3(3.0) | 57.8(4.1) | 9.2(4.3) | 19.8(9.8) | 8.0(5.6) |
| Outer regional/remote | 0.7(0.2) | 4.8(2.0) | 80.4(7.9) | 1.4(1.8) | 55.8(5.47) | 11.0(4.2) | 22.6(10.4) | 11.4(6.5) |
| Total | 0.8(0.2) | 4.2(3.4) | 83.1(10.6) | 4.5(3.7) | 56.4(5.6) | 9.3(3.6) | 22.3(9.3) | 8.0(5.5) |
| Test for trend | p < .001 | 0.100 | 0.007 | p < .001 | 0.858 | 0.008 | 0.462 | p < .001 |
| **Area disadvantage** | | | | | | | | |
| Most disadvantaged | 0.8(0.2) | 2.7(2.6) | 88.2(9.7) | 4.8(4.9) | 52.5(3.6) | 10.3(4.0) | 26.0(6.7) | 9.3(6.5) |
| 2 | 0.8(0.2) | 3.6(2.5) | 84.8(9.7) | 3.1(3.8) | 54.3(4.0) | 10.3(3.6) | 24.7(9.8) | 10.2(6.0) |
| 3 | 0.8(0.2) | 4.5(2.5) | 80.9(8.9) | 3.6(2.8) | 56.9(4.5) | 10.6(2.5) | 23.8(9.4) | 6.2(5.2) |
| 4 | 0.9(0.3) | 3.9(3.4) | 84.6(11.4) | 5.0(2.5) | 56.1(5.8) | 7.6(2.5) | 21.0(7.9) | 7.2(3.2) |
| Least disadvantaged | 0.9(0.2) | 6.0(4.8) | 77.4(10.9) | 6.2(3.1) | 62.2(4.7) | 7.7(4.0) | 15.9(9.3) | 6.7(4.9) |
| Total | 0.8(0.2) | 4.2(3.4) | 83.1(10.6) | 4.5(3.7) | 56.4(5.6) | 9.3(3.6) | 22.3(9.3) | 8.0(5.5) |
| Test for trend | 0.076 | 0.005 | 0.004 | 0.07 | p < .001 | 0.004 | p < .001 | 0.048 |

Abbrev. SD, standard deviation; PHC, primary health care; AIHW, Australian Institute of Health and Welfare; FTE, full-time equivalent; OOP, out-of-pocket; CD, chronic disease; HA, health assessment. Community health centres per 10,000 usual resident population. Test for trend for bivariate analyses, ordinary least squares regression.

characteristics from other domains (Table 4). For example, areas that were more affordable (that is, lower OOP expenses or more bulk-billing) also had more after-hours care (OOP expenses r = -0.42, bulk-billing r = 0.51, p < .001). Affordability also had a strong relationship with comprehensiveness, but the direction of association depended on the measure examined. That is, areas that were more affordable had more chronic disease care (OOP expenses r = -0.4, p < .001, bulk-billing r = 0.52, p<0.001), but less cervical screening (OOP expenses r = 0.55, p < .001, bulk-billing r = -0.64, p<0.001), as shown for bulk-billing in Fig 3.

When stratified by region, areas in major cities that had more GPs FTE per capita were *less* affordable (OOP expenses r = 0.45, p<0.01, bulk-billing r = -0.41, p < .01, S1 Appendix) and there was no correlation between availability and measures of comprehensiveness across all categories of remoteness—in particular the negative association with community health centres did not persist.

## Discussion

This study has, for the first time in Australia, constructed a suite of small-area measures of service organisation and delivery that reflect the core dimensions of effective PHC systems and represent avenues for policy and service delivery change. A total of thirteen variables were constructed; nine related to access (four relating to availability, four relating to affordability, one relating to accommodation), four capturing comprehensiveness and two capturing coordination. Our study provides further evidence that the organisation and delivery of services varies not only by remoteness of location, but also geographically at the small-area level.

Our finding that considerable geographic variation by small area exists in service-level characteristics of PHC, particularly in relation to remoteness and area socioeconomic disadvantage, was generally consistent with the literature [5–7, 10, 26–28, 36]. Consistent with previous studies, we found that disadvantaged metropolitan areas are the most affordable in terms of care [7], and provide more chronic disease care [8] that, as shown by our study, is relative to the population in need of this. These findings confirm an important and ongoing source of

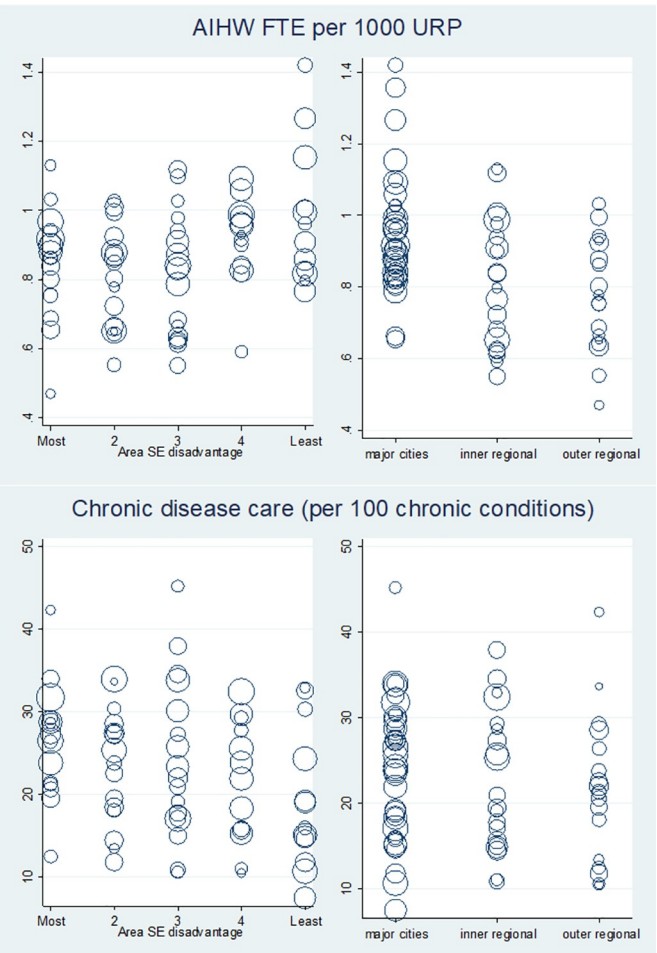

**Fig 2. Scatterplots of region and area disadvantage with area PHC service characteristic measures–Availability and comprehensiveness.** Abbrev. SE, socioeconomic; AIHW, Australian Institute of health and Welfare; FTE, full-time equivalent; URP, usual resident population; %, percentage: Size of circle is proportional to population of the SA3 plotted; that is, a larger circle indicates a larger population in that SA3.

equity in the Australian PHC system. Likewise, regional and remote areas also provide more health assessments [6] as do disadvantaged areas [10]. On the other hand, our study also found that major cities have the most availability [26, 27, 37], and advantaged areas in major cities even more so, representing a potential source of inequity in the health system. Similarly, while higher rates of cervical screening in advantaged areas have been previously reported [38], our findings clarify this is the case in all regions. In terms of equity, the findings suggest that policy initiatives should focus on the maldistribution of availability, not only by remoteness but also area disadvantage, and improving preventative care in disadvantaged areas across all regions.

Our finding of lower rates of MBS after-hours services in regional and remote areas, while consistent with a previous national report [8], contrasts with findings from a self-reported survey of after-hours care provided by GPs [36]. Given that much after-hours care in regional areas is provided within local hospitals, with different sources of funding, a large proportion of the after-hours services provided in regional and remote areas will not be captured by the MBS data used in this study. This most likely accounts for difference in findings between the two studies. Nevertheless, our finding that disadvantaged areas in inner regional locations had less

**Table 3. Association of area PHC service characteristics with area disadvantage, by region, mean (SD).**

| Major cities | | | | | | | | |
|---|---|---|---|---|---|---|---|---|
| Area SE disadvantage | GP FTE per capita | OOP expenses | Bulk-billing | After-hours care | HA | Cervical screening | CD care | Community health centres |
| 1 (most disadvantaged) | 0.9(0.1) | 0.3(0.3) | 97.4 (1.6) | 10.3(3.0) | 7.9(2.2) | 50.7(3.8) | 28.7(3.4) | 6.1(2.7) |
| 2 | 0.9(0.1) | 0.5(0.4) | 96.2(2.8) | 7.9(3.4) | 7.2(1.4) | 50.2(2.0) | 27.5(5.7) | 8.1(1.6) |
| 3 | 0.9(0.1) | 2.5(2.3) | 87.5(9.1) | 6.3(1.5) | 10.8(1.8) | 55.7(5.4) | 29.4(9.6) | 3.6(2.9) |
| 4 | 1.0(0.2) | 2.8(2.3) | 90.0(4.9) | 6.3(1.9) | 8.5(1.9) | 53.5(3.0) | 23.1(5.6) | 5.9(2.6) |
| 5. (least disadvantaged) | 1.0(0.2) | 7.4(4.9) | 75.7(11.3) | 6.2(2.9) | 8.2(2.2) | 62.5(4.9) | 17.6(8.0) | 6.6(5.5) |
| Total | 0.9(0.2) | 3.5(4.2) | 87.0(11.5) | 7.2(2.9) | 8.5(2.2) | 55.9(6.4) | 23.8 (8.2) | 6.1(3.8) |
| Test for trend | 0.037 | < .001 | < .001 | 0.003 | 0.715 | < .001 | < .001 | 0.987 |
| **Inner regional** | | | | | | | | |
| Area SE disadvantage | GP FTE per capita | OOP expenses | Bulk-billing | After-hours care | HA | Cervical screening | CD care | Community health centres |
| 1 (most disadvantaged) | 1.0(0.2) | 4.8(3.4) | 79.5(13.1) | 1.3(0.7) | 11.5(2.6) | 55.1(2.6) | 24.0(6.5) | 7.2(0.3) |
| 2 | 0.8(0.2) | 4.1(1.5) | 81.8(5.8) | 1.7(2.2) | 11.4(3.1) | 55.9(2.8) | 24.4(7.0) | 9.6(9.7) |
| 3 | 0.8(0.2) | 5.3(2.2) | 78.2(7.7) | 2.5(2.4) | 10.3(3.1) | 57.4(4.5) | 22.0(8.5) | 7.6(6.3) |
| 4 | 0.8(0.2) | 5.3(4.8) | 77.6(15.2) | 3.5(2.4) | 7.2(3.2) | 56.8(1.8) | 19.4(10.8) | 8.7(3.4) |
| 5. (least disadvantaged) | 0.8(0.1) | 3.0(2.8) | 81.7(9.5) | 6.1(3.9) | 6.7(6.6) | 61.3(4.7) | 12.2(11.6) | 7.1(1.4) |
| Total | 0.8(0.2) | 4.6(2.9) | 79.4(9.3) | 3.3(3.0) | 9.2(4.3) | 57.8(4.1) | 19.8(9.8) | 8.1(5.6) |
| Test for trend | 0.413 | 0.425 | 0.944 | 0.006 | 0.026 | 0.035 | 0.029 | 0.868 |
| **Outer regional remote** | | | | | | | | |
| Area SE disadvantage | GP FTE per capita | OOP expenses | Bulk-billing | After-hours care | HA | Cervical screening | CD care | Community health centres |
| 1 (most disadvantaged) | 0.8(0.2) | 4.1(2.0) | 83.0(7.3) | 1.3(0.9) | 11.8(4.7) | 53.1(3.2) | 24.5(8.5) | 12.2(8.0) |
| 2 | 0.7(0.2) | 5.0(1.9) | 80.2(8.6) | 1.2(2.2) | 11.5(3.8) | 55.7(3.9) | 23.4(12.4) | 11.6(6.0) |
| 3 | 0.6(0.0) | 6.4(0.6) | 74.9(0.6) | 1.0(0.3) | 11.4(2.5) | 57.8(2.4) | 16.3(8.1) | 7.0(1.7) |
| 4 | 0.5(0.7) | 5.2(3.5) | 71(-) | 3.5(3.7) | 4.9(1.4) | 66.2(13.7) | 10.5(-) | 10.7(-) |
| 5. (least disadvantaged) | – | – | – | – | – | – | – | – |
| Total | 0.7(0.2) | 4.8(2.0) | 80.4(7.9) | 1.4(1.8) | 11.0(4.2) | 55.8(5.7) | 22.6(10.4) | 11.4(6.5) |
| Test for trend | 0.09 | 0.214 | 0.07 | 0.291 | 0.094 | 0.002 | 0.173 | 0.481 |

Abbrev: CD, chronic disease; CHC, community health centre; FTE, full-time equivalent; GP, general practitioner; HA, health assessment; PHC, primary health care; OOP, out-of-pocket; SD, standard deviation; SE, socioeconomic. FTE per 1000 URP. Community health centres per 10,0000 URP. Test for trend bivariate analyses OLS regression.

**Table 4. Correlation between dimensions of area PHC service characteristics (including components of access).**

| Dimension/access component | PHC characteristic | Availability | | Affordability | | | | Accommodation | |
|---|---|---|---|---|---|---|---|---|---|
| | | GP FTE per capita | | OOP expenses | | Bulk-billing | | After-hours care | |
| | | r | p | r | p | r | p | r | p |
| **Affordability** | OOP expenses | 0.13 | 0.206 | – | – | – | – | – | – |
| | Bulk-billing | -0.03 | 0.789 | – | – | – | – | – | – |
| **Accommodation** | After-hours care | 0.19 | 0.08 | -0.42 | < .001 | 0.51 | < .001 | – | – |
| **Comprehensive- ness/ coordination** | Health assessments | -0.02 | 0.854 | -0.05 | 0.641 | 0.09 | 0.425 | -0.22 | 0.041 |
| | Cervical screening | 0.07 | 0.505 | 0.55 | < .001 | -0.64 | < .001 | -0.12 | 0.268 |
| | CD care | 0.12 | 0.274 | -0.40 | < .001 | 0.52 | < .001 | 0.17 | 0.117 |
| | Community health centres | -0.20 | 0.039 | 0.04 | 0.717 | -0.11 | 0.308 | -0.20 | 0.003 |

Abbrev. GP, general practitioner, PHC, primary health care; FTE; full time equivalent; OOP, out-of-pocket; CD, chronic disease; Pearson's correlation coefficients; p, p-value. Per capita, per 1000 usual resident population (URP). Community health centres per 10,000 URP.

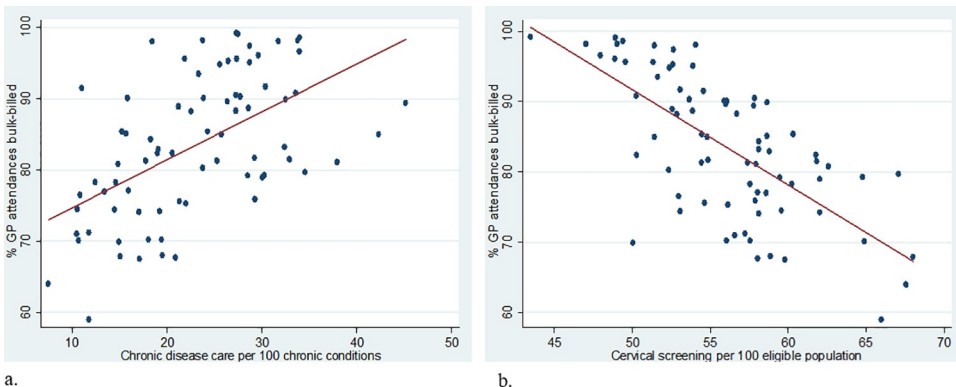

**Fig 3.** Relationship between dimensions of area PHC service characteristics: Scatterplot of bulk-billing with (a) CD care and (b) cervical screening. Abbrev. PHC primary health care; GP, general practice; CD, chronic disease. Eligible population for cervical screening is percentage of women aged 20–69 who have participated in cervical screening. Red line represents line of best fit.

after-hours care points to a further potential source of inequity that could be redressed through policy.

The relationship between dimensions of PHC service delivery at the area-level differed according to which measures were examined. The inverse relationship between availability and affordability has been previously demonstrated in Australia [39]. This contrasts with international studies [15, 40], likely due to differences in overall numbers of primary care physicians and ability to access specialty care directly, as is common in the US and some European countries. The finding that areas that were more affordable also had more after-hours care is also consistent with previous Australian [36] and international studies [15, 41]. While not previously examined in Australia, the general finding of significant correlation between characteristics from the dimensions of access (availability, affordability and accommodation) and comprehensiveness and coordination, is consistent with international studies [2, 15, 18, 42, 43].

## Strengths and limitations

An important strength of this study is that data were collated from a range of routine data sources available at a geographic scale most relevant to the organisation and delivery of PHC in Australia. Further, this is the first Australian study to construct small-area measures that encompass the core dimensions at the service delivery level of PHC systems; that is, access (and its components), comprehensiveness and coordination. All numerator data are full census counts (rather than a sample) for each variable in each area, increasing the accuracy of representation of these characteristics. Further, denominator data for comprehensiveness and coordination accounted for the eligible population, rather than total population as seen with previously published figures. This provides a better estimate of services available relative to overall need in an area, thereby improving face validity and interpretation of findings when applying these variables in analyses.

However, there are important data limitations that require consideration. There are potential inaccuracies in the main availability estimate (AIHW FTE) related to data suppression due to confidentiality concerns, primarily affecting outer regional, remote or very remote areas. Given the few instances in which this occurred, the overall effect is likely negligible. The measures pertaining to comprehensiveness and coordination are approximations developed with the best available data at this geographical scale. Ideally, measures would reflect the extent to

which most services in an area incorporate aspects of coordination (for example, team-based care, role substitution, skill-mix) and comprehensiveness (for example, specific programs for chronic diseases or maternal/child health, health promotion activities, scope of practice scores for providers) rather than the output that such resources and activities enable (for example, claiming chronic disease items). Data of this richness are currently unavailable. Further, data relating to other aspects of accommodation such as appointment systems, walk-in facilities, telephone and email services are not routinely available. However, with the mandate on Primary Health Networks (Australia's regional bodies responsible for coordinating primary healthcare services) to undertake needs assessment and identify service gaps [44], and to report on this, there is the potential for such data to be collected at a geographically useful scale. There are promising emerging efforts to that end [45–50].

Further in relation to comprehensiveness and coordination measures, we found that the highest numbers of community health centres relative to the population were observed in remote and disadvantaged areas, and this measure was poorly correlated with other measures within the comprehensiveness domain and negatively associated with availability. This suggests that the community health centre measure may be measuring availability of supporting or supplementary services. The positive association of cervical screening with advantaged areas and negative relationship with affordability and other measures of comprehensiveness suggests that this may better reflect health behaviour (e.g. propensity to seek care and capacity to overcome opportunity costs) and consequent care received within that area, rather than an area PHC service characteristic per se.

## Conclusion

Identifying avenues for PHC system reform requires appropriate empirical measurement of the organisation and delivery of services of areas. The extent and nature of how this varies may then provide insights as to best practice for achieving equitable and high-quality care. To that end, this study offers direction and clarification. Given the available data, the measures constructed represent the best approximation at a meaningful geographical scale from the domains of access, comprehensiveness and coordination relevant to policy and service planning. In terms of equity, initiatives should consider addressing the maldistribution of GPs by remoteness and area disadvantage. Initiatives should also consider increasing preventative care in disadvantaged areas across all regions and after-hours care in disadvantaged regional locations.

## Supporting information

**S1 Appendix. List of MBS item numbers included and supplementary tables.**
(DOCX)

## Author Contributions

**Conceptualization:** D. C. Butler, L. R. Jorm, S. Larkins, J. Humphreys, K. J. Korda.

**Formal analysis:** D. C. Butler.

**Methodology:** D. C. Butler, K. J. Korda.

**Supervision:** L. R. Jorm, S. Larkins, J. Humphreys, K. J. Korda.

**Writing – original draft:** D. C. Butler.

**Writing – review & editing:** D. C. Butler, L. R. Jorm, S. Larkins, J. Humphreys, J. Desborough, K. J. Korda.

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
