## [Decision Letter · Decision Letter 0]

12 Oct 2021

PONE-D-21-15234Examining area-level variation in service organisation and delivery across the breadth of primary healthcare. Usefulness of measures constructed from routine dataPLOS ONE

Dear Dr. Butler,

Thank you for submitting your manuscript to PLOS ONE. After careful consideration, we feel that it has merit but does not fully meet PLOS ONE’s publication criteria as it currently stands. Therefore, we invite you to submit a revised version of the manuscript that addresses the points raised during the review process.

ACADEMIC EDITOR: Considering my own reading and reviewer suggestion, I am recommending a minor revision for this manuscript. Please address the minor comments made by reviewer and submit the revision. 

We look forward to receiving your revised manuscript.

Kind regards,

Srinivas Goli, Ph.D.

Academic Editor

PLOS ONE

a) Did participants provide their written or verbal informed consent to participate in this study?

“This research was supported through a grant from the Australian Government through the National Health and Medical Research Council Postgraduate Scholarship (GNT1038903).”

“DB. This research was supported through a grant from the Australian Government through the National Health and Medical Research Council Postgraduate Scholarship (GNT1038903). The funders had no role in study design, data collection and analysis, decision to publish, or preparation of the manuscript.”

5. We note that you have included the phrase “data not shown” or “data not available” in your manuscript. Unfortunately, this does not meet our data sharing requirements. PLOS does not permit references to inaccessible data. We require that authors provide all relevant data within the paper, Supporting Information files, or in an acceptable, public repository. Please add a citation to support this phrase or upload the data that corresponds with these findings to a stable repository (such as Figshare or Dryad) and provide and URLs, DOIs, or accession numbers that may be used to access these data. Or, if the data are not a core part of the research being presented in your study, we ask that you remove the phrase that refers to these data.

Additional Editor Comments (if provided):

Considering my own reading and reviewer suggestion, I am recommending a minor revision for this manuscript. Please address the minor comments made by reviewer and submit the revision.

Reviewers' comments:

Reviewer's Responses to Questions

**Comments to the Author**

1. Is the manuscript technically sound, and do the data support the conclusions?

Reviewer #1: Yes

2. Has the statistical analysis been performed appropriately and rigorously? 

Reviewer #1: Yes

3. Have the authors made all data underlying the findings in their manuscript fully available?

Reviewer #1: Yes

4. Is the manuscript presented in an intelligible fashion and written in standard English?

Reviewer #1: Yes

5. Review Comments to the Author

Reviewer #1: -Under conclusion the authors may suggest a targeted policy recommendation based one the major cities and inner cities variations where there are gaps in service delivery.

-Under the map output authors should provide corresponding key interpretation to colour labelling of “1 2 3 4”

6. PLOS authors have the option to publish the peer review history of their article (what does this mean?). If published, this will include your full peer review and any attached files.

Reviewer #1: **Yes: **Kofi Aduo-Adjei

---

## [Author Response · Author response to Decision Letter 0]

21 Oct 2021

21st October 2021

Professor Tanya Doherty

Guest Editor, PLOS One call for Papers Health Services Research 

Dear Professor Doherty,

TITLE Examining area-level variation in service organisation and delivery across the breadth of primary healthcare. Usefulness of measures constructed from routine data 

We wish to thank you for the opportunity to resubmit a revised version of our paper publication as original research in PLOS One call for Papers Health Services Research having addressed the additional requirements outlined by the Editor and the reviewer.

Please see our detailed numbered responses below.

RESPONSE TO THE EDITOR.

1. Please ensure that your manuscript meets PLOS ONE's style requirements, including those for file naming

RESPONSE: We have reviewed our manuscript in detail and can confirm that it meets PLOS ONE’s style requirement as outlined in the templates available at the links provided

a) Did participants provide their written or verbal informed consent to participate in this study?

RESPONSE: Thank you for bringing this to our attention. We request our ethics statement to be amended as follows:

Approval for this project was obtained from the Australian National University Human Research Ethics Committee (protocol 2011/703) and the AIHW Ethics Committee (EC2010/2/23). As this was a secondary analysis of aggregated data, individual consent (verbal or written) was not required.

“This research was supported through a grant from the Australian Government through the National Health and Medical Research Council Postgraduate Scholarship (GNT1038903).”

RESPONSE: Thank you for bringing this to our attention. We have deleted the acknowledgement section from the manuscript.

“DB. This research was supported through a grant from the Australian Government through the National Health and Medical Research Council Postgraduate Scholarship (GNT1038903). The funders had no role in study design, data collection and analysis, decision to publish, or preparation of the manuscript.”

RESPONSE: Thank you for this clarification. Having deleted the funding statement from the acknowledgement section, we wish for funding statement to remain in its current form.

RESPONSE: Thank you for bringing this to our attention and your clarification. We wish for our data availability statement to be amended as follows.

ABS Census and PHIDU data are available for download at https://www.abs.gov.au/websitedbs/censushome.nsf/home/historicaldata2006 and https://phidu.torrens.edu.au/social-health-atlases respectively. Other data used in this study cannot be shared publicly because data are owned by a third party and authors do not have permission to share these data. Detailed information for accessing the data underlying the results presented in the study are available from: AIHW health workforce, https://www.aihw.gov.au/about-our-data/accessing-data-through-the-aihw/data-on-request; AMPCo Medical workforce, https://www.ampco.com.au/ampco-data-services/; HERO data, http://www.healthstats.nsw.gov.au; MBS claims, https://www.servicesaustralia.gov.au/organisations/about-us/reports-and-statistics/statistical-information-and-data#contacts; and ARIA + https://arts.adelaide.edu.au/hugo-centre/services/aria.

5. We note that you have included the phrase “data not shown” or “data not available” in your manuscript. Unfortunately, this does not meet our data sharing requirements. PLOS does not permit references to inaccessible data. We require that authors provide all relevant data within the paper, Supporting Information files, or in an acceptable, public repository. Please add a citation to support this phrase or upload the data that corresponds with these findings to a stable repository (such as Figshare or Dryad) and provide and URLs, DOIs, or accession numbers that may be used to access these data. Or, if the data are not a core part of the research being presented in your study, we ask that you remove the phrase that refers to these data

RESPONSE: On review of the manuscript the only section where this phrase is used is as follows:

“Data not available at SA3 level were re-assigned using relevant population weighted correspondences publicly available through the ABS (that is, from 2006 statistical geographies to 2011 statistical geographies).” (methods, second last paragraph)

This is not referring to data or results being withheld, but rather indicating how we aggregated the data for analysis. As such we have not made changes to the manuscript in relation to this. 

RESPONSE: We can confirm that our ethics statement appears only in the methods section of our manuscript.

RESPONSE: We have reviewed our reference list and can confirm it is complete and correct.

RESPONSE TO THE REVIEWER.

1. Reviewer #1: -Under conclusion the authors may suggest a targeted policy recommendation based one the major cities and inner cities variations where there are gaps in service delivery.

-Under the map output authors should provide corresponding key interpretation to colour labelling of “1 2 3 4”

RESPONSE: We thank the reviewer for their comments. We have revised the conclusion in line with the reviewer’s suggestion to the following.

Identifying avenues for PHC system reform requires appropriate empirical measurement of the organisation and delivery of services of areas. The extent and nature of how this varies may then provide insights as to best practice for achieving equitable and high-quality care. To that end, this study offers direction and clarification. Given the available data, the measures constructed represent the best approximation at a meaningful geographical scale from the domains of access, comprehensiveness and coordination relevant to policy and service planning. In terms of equity, initiatives should consider addressing the maldistribution of GPs by remoteness and area disadvantage. Initiatives should also consider increasing preventative care in disadvantaged areas across all regions and after-hours care in disadvantaged regional locations.

With respect to the map output, the corresponding key interpretation to the colour labelling is provided in the figure legend. We defer to the editors preference as to whether this should be amended to be included in the image. 

We also wish to bring to the attention of the Editor that we have made some very minor edits for clarity of wording and to correct grammatical errors (see page 5, 17, 19).

We thank the Editor and reviewer for their time and consideration of our manuscript. We very much hope you will find it suitable for publication in PLOS One-call for papers, Health Services Research. 

Yours sincerely, 

Dr Danielle Butler, corresponding author, on behalf of all authors

---

## [Decision Letter · Decision Letter 1]

15 Nov 2021

Examining area-level variation in service organisation and delivery across the breadth of primary healthcare. Usefulness of measures constructed from routine data

PONE-D-21-15234R1

Dear Dr. Butler,

We’re pleased to inform you that your manuscript has been judged scientifically suitable for publication and will be formally accepted for publication once it meets all outstanding technical requirements.

Kind regards,

Srinivas Goli, Ph.D.

Academic Editor

PLOS ONE

Additional Editor Comments (optional):

Considering the reviewer suggestion and my own reading of this paper, I am recommending this paper for publication.

Reviewers' comments:

Reviewer's Responses to Questions

**Comments to the Author**

1. If the authors have adequately addressed your comments raised in a previous round of review and you feel that this manuscript is now acceptable for publication, you may indicate that here to bypass the “Comments to the Author” section, enter your conflict of interest statement in the “Confidential to Editor” section, and submit your "Accept" recommendation.

Reviewer #1: All comments have been addressed

2. Is the manuscript technically sound, and do the data support the conclusions?

Reviewer #1: Yes

3. Has the statistical analysis been performed appropriately and rigorously? 

Reviewer #1: Yes

4. Have the authors made all data underlying the findings in their manuscript fully available?

Reviewer #1: Yes

5. Is the manuscript presented in an intelligible fashion and written in standard English?

Reviewer #1: Yes

6. Review Comments to the Author

Reviewer #1: (No Response)

7. PLOS authors have the option to publish the peer review history of their article (what does this mean?). If published, this will include your full peer review and any attached files.

Reviewer #1: **Yes: **Kofi Aduo-Adjei

---

## [Editor Report · Acceptance letter]

18 Nov 2021

PONE-D-21-15234R1 

Examining area-level variation in service organisation and delivery across the breadth of primary healthcare. Usefulness of measures constructed from routine data 

Dear Dr. Butler:

I'm pleased to inform you that your manuscript has been deemed suitable for publication in PLOS ONE. Congratulations! Your manuscript is now with our production department. 

Kind regards, 

on behalf of

Dr. Srinivas Goli 

Academic Editor

PLOS ONE